# Genomic Characterization of Three Novel *Bartonella* Strains in a Rodent and Two Bat Species from Mexico

**DOI:** 10.3390/microorganisms11020340

**Published:** 2023-01-30

**Authors:** Jonathan Gonçalves-Oliveira, Ricardo Gutierrez, Cory Lee Schlesener, David A. Jaffe, Alvaro Aguilar-Setién, Henri-Jean Boulouis, Yaarit Nachum-Biala, Bihua C. Huang, Bart C. Weimer, Bruno B. Chomel, Shimon Harrus

**Affiliations:** 1Laboratory of Zoonotic Vector-Borne Diseases—The Koret School of Veterinary Medicine, The Hebrew University of Jerusalem, Rehovot 76100, Israel; 2National Reference Center for Bacteriology, Costa Rican Institute for Research and Teaching in Nutrition and Health (INCIENSA), Tres Rios 4-2250, Costa Rica; 3Department of Population Health and Reproduction, 100K Pathogen Genome Project, School of Veterinary Medicine, University of California, Davis, CA 95616, USA; 4Unidad de Investigación Médica en Inmunología, Hospital de Pediatría, Centro Médico Nacional Siglo XXI, IMSS, Mexico City 06720, Mexico; 5Équipe Vecteurs et Agents Microbiens Pathogènes ENVA–Anses–INRA–UMR BIPAR, École Nationale Vétérinaire d’Alfort, 7, Avenue du Général de Gaulle, Maisons-Alfort, CEDEX, 94704 Paris, France

**Keywords:** *Bartonella*, genomes, rodents, bats, Mexico

## Abstract

Rodents and bats are the most diverse mammal group that host *Bartonella* species. In the Americas, they were described as harboring *Bartonella* species; however, they were mostly characterized to the genotypic level. We describe here *Bartonella* isolates obtained from blood samples of one rodent (*Peromyscus yucatanicus* from San José Pibtuch, Yucatan) and two bat species (*Desmodus rotundus* from Progreso, and *Pteronotus parnellii* from Chamela-Cuitzmala) from Mexico. We sequenced and described the genomic features of three *Bartonella* strains and performed phylogenomic and pangenome analyses to decipher their phylogenetic relationships. The mouse-associated genome was closely related to *Bartonella vinsonii*. The two bat-associated genomes clustered into a single distinct clade in between lineages 3 and 4, suggesting to be an ancestor of the rodent-associated *Bartonella* clade (lineage 4). These three genomes showed <95% OrthoANI values compared to any other *Bartonella* genome, and therefore should be considered as novel species. In addition, our analyses suggest that the *B. vinsonii* complex should be revised, and all *B. vinsonii* subspecies need to be renamed and considered as full species. The phylogenomic clustering of the bat-associated *Bartonella* strains and their virulence factor profile (lack of the Vbh/TraG conjugation system remains of the T4SS) suggest that it should be considered as a new lineage clade (L5) within the *Bartonella* genus.

## 1. Introduction

*Bartonella* species are Gram-negative, facultative intracellular bacteria, transmitted by arthropods, infecting mammalian erythrocytes and endothelial cells [1]. Currently, there are 39 species and three subspecies with a valid taxonomic standing in the genus, at least 20 of which have been associated with human diseases [2,3]. As the greatest diversity of *Bartonella* species is reported in rodents and bats [4,5,6], rodents and possibly bats are considered as a potential source of emerging zoonotic *Bartonella* pathogens [7]. Moreover, bats were reported to have a crucial role in the diversification of the *Bartonella* genus [8].

Several studies described the presence of *Bartonella* DNA in samples of rodents and bats in the Americas [5,9,10,11,12,13,14,15]. However, the genomic diversity of *Bartonella* species detected or isolated in mammal hosts, especially in the Americas has not been fully described and has yet to be investigated and elucidated [14,15]. Therefore, this study describes three novel *Bartonella* genomes acquired from Mexican rodents and bats. We analyzed their phylogenomic relationships and genome divergence among all known *Bartonella* species.

## 2. Methods

### 2.1. Sampling and Isolation

Blood samples were obtained from three different sites in Mexico in previous studies [14,15]. A rodent sample (strain 220B) was collected from San José Pibtuch, Yucatan and two bat samples (strains F02 and A05) were from Chamela-Cuitzmala (F02) and Progreso (A05), respectively. Strain 220B was isolated from the blood of a Yucatan deer mouse (*Peromyscus yucatanicus*). Strains A05 and F02 were isolated from the blood of a common vampire (*Desmodus rotundus*) and a Parnell mustached bat (*Pteronotus parnellii*), respectively. The blood samples from these three strains were cultured on chocolate agar plates using 200 µL blood and M199 growth media with 5% amphotericin B. Suspected *Bartonella* colonies were re-plated and the DNA was extracted and submitted to conventional PCR to confirm that they belong to the *Bartonella* genus. Target genes of *Bartonella* from these samples are available in GenBank (*gltA*: MN394838, KY629834, KY629884; *ftsZ*: KY629824, KY629797; *rpoB*: KY629925, KY629898). After confirmation as *Bartonella*, the three samples were subjected to WGS.

### 2.2. Genomic and Library Methods

The genomic DNA extraction was performed following growth on chocolate agar. Isolates were selected and the gDNA was extracted after lysis using Qiagen DNeasy blood and tissue kits [14]. Each sample was used to produce a sequencing library with gDNA that was of sufficient quality (A_260/280_ and A_260/230_ > 1.5) with an average insert size of 350 bp according to the KAPA HyperPlus Kit protocol as described previously [16,17,18,19]. All samples were sequenced using HiSeq XTEN Sequencing System (Illumina, San Diego, CA, USA) [20,21].

### 2.3. Genome Analysis

The whole genome sequences were trimmed, quality filtered, and quality assured using TrimGalore (v.0.5.0) [22]. The trimmed reads were submitted to the comprehensive genome analysis in the PATRIC database using UNICYCLER v.0.4.8 [23]. The resulting assembly was analyzed in PATRIC [24] and MiGa [25] to evaluate the genome features and completeness, contamination, and quality. The genomes were annotated by Bakta (v.1.6.1) [26], Prokka (v.3.2.1) [27], RASTtk (v1.073) [28], as well as PGAP (v.6.4) [29], and the PGAP was chosen as the reference annotation. A whole-genome-based phylogenetic tree was reconstructed using RAxML (v8.2.11) [30] implemented in PATRIC [31]. Whole-genome average nucleotide identity (ANI) was calculated between genomes of members of the genus *Bartonella* using the software Orthologous Average Nucleotide Identity Tool (OAT v.0.93.1) [18,32]. Delimitation of species using OrthoANI results was based on Goris et al. [33] and Ciufo et al. [34], considering <95% values as novel species [19].

### 2.4. Virulence Factors Characterization

The *Bartonella* lineages were classified according to their virulence factors (BaGTA, Flagellum, TrW T4SS, Vbht T4SS, VirB/D4 T4SS) following Wagner and Dehio [35]. The sequences and annotations of all *Bartonella* genomes were retrieved from the NCBI database and analyzed by Geneious program v.7 (Biomatters, Newark, NJ, USA). All the cassettes of virulence factor genes found in their genomes were evaluated and indicated on a phylogenetic tree.

### 2.5. Pangenome Analysis

The pangenome of the *Bartonella* genus was determined by the same genome sequences used in other analyses, including the *Brucella abortus* as an outgroup. The quality of genome sequences, contamination, completeness, and gene markers of lineage was performed using the CheckM [36]. The Prokka annotation was used as an input for pangenome analysis using Roary v.3.12.0 [37]. Core genes were considered to be those present in >90% of the genomes in the comparison [38]. To evaluate the similarity between the gene sequences and compare shared sequences in the data, the sourmash v.3.2.3 was used [39]. A neighbor-joining phylogeny reconstruction was built using the sourmash matrix output of shared coding gene [40]. The “roary_plots.py” script was used to visualize a matrix with gene presence/absence of core, shell, and accessory genes [41]. Gene presence/absence and the neighbor-joining phylogenetic reconstruction were visualized using Phandango [42].

## 3. Results

*Bartonella* isolates were sequenced and their genome features are described in Table 1. Two genomes from bat samples were considered as complete genomes (strains F02 and A05), with one hundred and six essential genes and one nearly complete genome with one hundred and five essential genes, with only the ribosomal protein L34 gene missing (strain 220B) (Table 1). This Whole Genome Shotgun project has been deposited at DDB/ENA/GenBank under accession numbers: JANHOI000000000 (strain 220B); JANHOJ000000000 (strain A05); JANHOK000000000 (strain F02); Bioproject PRJNA8621808; BioSample: SAMN29927589-SAMN29927591.

The topology of 137,193 nucleotides from 149 coding genes shows high nodal support (>95 bootstrap, Figure 1) and indicates a diversification of these isolates into two clades. The first is composed of two new strains, *Bartonella* strains A05 and F02 from bats of the species *D. rotundus* and *P. parnellii*, respectively. The topology of these two strains appears as a single branch of lineage 4 located between lineages 3 and 4 suggesting that it is an ancestor clade of the rodent-associated *Bartonella* (Figure 1). The strain 220B from the mouse *Peromyscus yucatanicus* is clustered as a sister group with the complex *B. vinsonii* in lineage 4 clade (Figure 1).

The OrthoANI between strains A05 and F02 was 79.8%. The strain A05 had OrthoANI values between 78.6–80.07% with the species *B. alsatica, B. vinsonii* subsp. *vinsonii*, *B. taylorii*, *B. henselae*, *B. quintana*, *B. doshiae, B. tribocorum,* and *B. grahamii* (Figure 2). The strain F02 had OrthoANI values between 78.2–78.8% with the species *B. alsatica, B. vinsonii* subsp. *vinsonii*, *B. taylorii*, *B. henselae*, *B. quintana*, *B. doshiae, B. tribocorum,* and *B. grahamii* (Figure 2). The strain 220B had OrthoANI values of 90–91% with the *B. vinsonii* complex, 85% with *B. taylorii,* 84% with *B. alsatica* and *B. phoceensis*, 83.5% with *B. florencae,* and 82% with *B. rattaustraliani* (Figure 2).

Strain 220B contains the same repertoire of virulence factor genes presented in lineage 4 (Figure 1). The annotation shows the virulence factor genes encoding the BaGTA, TrW T4SS, Vbh/TraG T4SS, Virb/D4 T4SS, and Beps. Two cassettes of complex VirB2-11 were found in this strain. Strain F2 contains genes of the BaGTA, TrW T4SS, Virb/D4 T4SS, and Beps; however, it does not contain genes of toxin Vbh/TraG T4SS. Two cassettes of complex VirB2-11 were detected in this strain. Strain A5 contains the BaGTA, TrW T4SS, and Virb/D4 T4SS virulence factors; however, the cassettes of genes encoding VBh/TraG T4SS and Beps were not identified.

*Bartonella* pangenome containing 39,431 genes and a core of 27 genes (>99%) and 81 soft core genes (95–99%) was observed for 60 genomes. The known lineages (1–4) were recovered from the 1668 genes (15–95%). The unique genes were the most representative of the genus with 37,655 genes. Strain 220B contains more genes shared with the *B. vinsonii* complex and with the rodent-associated *Bartonella* clade. The two bat-associated strains, A05 and F02, do not share the same cassettes of shell genes shared with other lineages. The pangenome analyses show that the *Bartonella* species are significantly different among each other (Appendix A).

## 4. Discussion

The genus *Bartonella* includes a great diversity of species harbored by many mammalian hosts worldwide. The genetic diversity of this genus has been described by comparisons of the genetic divergence and multilocus analyses of target genes, which allowed for the description of several species and genotypes [43,44]. In this study, we characterized three strains of *Bartonella* by whole genome sequencing (WGS) and according to Goris et al. [33], the *Bartonella* genome analysis and their OrthoANI values (cut-off less than <95%), they should be considered as three novel species of the genus *Bartonella*. However, to fully describe and name these bacteria as new species according to the International Code of Nomenclature of Prokaryotes (ICNP), the characterization of several phenotypic, biological, and biochemical aspects of the bacteria must be carried out in addition to the isolation and WGS [45].

Fischedick et al. [15] described *Bartonella* genotypes from the Yucatan dear mouse (*P. yucatanicus*) based on target genes and named strain 220B as *Bartonella vinsonii* subsp. *yucatanensis*; however, according to our analysis, its OrthoANI values of 90–91% indicate that it is a distinct species within the *B. vinsonii* complex. Our results support recent findings described by Amaral et al. [46] suggesting that the *B. vinsonii* complex needs to be revised, as the subspecies already described and named in this complex have OrthoANI values less than 95% between them (see Figure 2). This is particularly true for *B. vinsonii* subsp. *berkhoffii* that has mainly a carnivore reservoir rather than a rodent reservoir as described for the other subspecies [47]. Considering the virulence factors, the mouse-associated strain 220B has the cassette genes of BaGTA, TrW T4SS, Vbh/TraG T4SS, VirB/D4, and Beps (Figure 1). This is the same repertoire of virulence factors found in lineage 4 of the *Bartonella* genus. This lineage is commonly found in rodents [7,48]. The host of *Bartonella* strain 220B belongs to the genus *Peromyscus*. This rodent genus is distributed across North and Central America and is considered a natural reservoir of zoonotic viral and bacterial pathogens, such as Hantavirus and *Borrelia burgdorferi* [49,50]. Three species of *Peromyscus* have already been found to host *Bartonella* DNA in northwestern, central, and eastern Mexico including *P. leucopus*, *P. maniculatus,* and *P. yucatanicus* [14,51].

Two bat-associated complete genomes were annotated herein and indicated that they are new species with high divergence across the *Bartonella* species. The first ever bat-associated *Bartonella* genome to be sequenced was named *Bartonella* sp. HY038. This genome was sequenced from bat faeces in China, and is the most ancient ancestor of all *Bartonella* species (see Figure 1). The two described bat-associated genomes herein were the first genomes sequenced from isolates of bat blood samples. In the comparison between *Bartonella* sp. HY038 and strains A5 and F2, OrthoANI values of 68% and 69%, respectively, were retrieved, indicating that the great diversity of bat-associated *Bartonella* genomes is still underestimated. The bat-associated *Bartonella* genomes clustered into a single clade, distinct from the L3 and L4 lineages (Figure 1). The L4 lineage is the most diverse and mostly represented by rodent-associated *Bartonella* species described to date, as described in the mouse-associated 220B strain in this study, while the L3 has only two species, *B. clarridgeiae* and *B. rochalimae* with a valid status, mainly associated with domestic and wild carnivores [52]. The two novel species described in this study reveal a new lineage of *Bartonella*. The bat-associated genomes shared the TrW T4SS, the genetic cassette acquired following the loss of the flagellum [35]; however, both strains do not have the genes encoding Vbh/traG conjugation systems (Figure 1). According to Harms and Dehio [53], these genes have an important role in the pathogenesis of *Bartonella* species, and the complete cassette of Vbh conjugation systems are found in the *Bartonella* chromosomes or plasmids [35]. Most of the *Bartonella* species contain VbhA toxin and TraG genes in their genomes, two homologous regions known as part of the VbhT T4SS system; however, these genes were not identified in any bat-associated genome to date. The bat-associated strains have the VirB-like genes (Figure 1). This type IV secretion system is responsible for the translocation of *Bartonella* effector proteins (Beps) in different niches of mammalian host cells [53]. The Beps were detected in strain F02, but not in strain A05. Due to this high divergence between the virulence factors in these new strains and lineage 4, and their phylogenomic clustering into a single clade, distinct from lineages 3 and 4, herein we propose these strains as representative of a new lineage, lineage 5, within the *Bartonella* genus (Figure 1). Bats form a diverse group as rodents, and act in the dispersion of *Bartonella* to other mammalian hosts, which consequently contribute to the adaptation and diversification of these bacteria [8]. *P. parnelli* is distributed throughout North America and northern South America. The strain F02 isolated in this insectivorous bat was described by Stuckey et al. [14], with eight out of thirty (26.7%) specimens reported to be positive for *Bartonella* DNA. Previously, only flies and ticks collected from this species have been described to contain *Bartonella* DNA, both in Mexico and French Guiana [54,55]. The vampire bat *D. rotundus* is widely distributed from southern North America to southern South America and is known to be one of the main natural hosts of the rabies virus [56]. Due to its exclusive diet of mammalian blood, the ecological interaction between this species and other domestic and food producing animals is one of the biggest concerns regarding the spillover of pathogens in Latin America [57]. In addition, this species is known to roost with other bat species and share vectors, and therefore potentially cause a spill over of pathogens to other bat species [58]. From these ecological and dietary aspects, *D. rotundus* is an important host of *Bartonella* spp., with a high prevalence and great diversity of genotypes reported from Central and South America [5,9,14,58,59,60]. Previous studies have reported that *Bartonella* genotypes found in vampire bats were mainly associated with lineage 2 (L2), the ruminant-associated *Bartonella* spp. [9,60]. However, herein we describe a new *Bartonella* strain isolated from *D. rotundus* which was not associated with L2, and belongs probably to a new lineage, distant from those described in previous studies.

According to the pangenome analysis that was carried out in this study, the *Bartonella* species showed high variability of unique genes compared with shared and core genes. The pangenome analysis indicated that the strains described herein should be considered as new species. Additionally, the genetic distance cut-off (OrthoANI values) and core genes similarity (%) should be established for *Bartonella*, as previously conducted for *Hungatella hathewayi*, a necessary step toward evaluating the intrageneric diversity [18].

## 5. Conclusions

A deeper understanding of the evolutionary diversification observed in this genus is made possible by the characterization and description of additional genomes in rodents and bats. Our study using genomic approach shows that in Mexico, there are at least three novel *Bartonella* species described herein that need further characterization in order to be fully named according to the ICNP. Based on phylogenomic clustering and virulence factors, the bat-associated *Bartonella* species, presented in this study, suggest the possibility of a new fifth lineage in the genus. Additionally, the analysis of OrthoANI values carried out herein suggests that there is a need to revise the *B. vinsonii* complex.

## Figures and Tables

**Figure 1 microorganisms-11-00340-f001:**
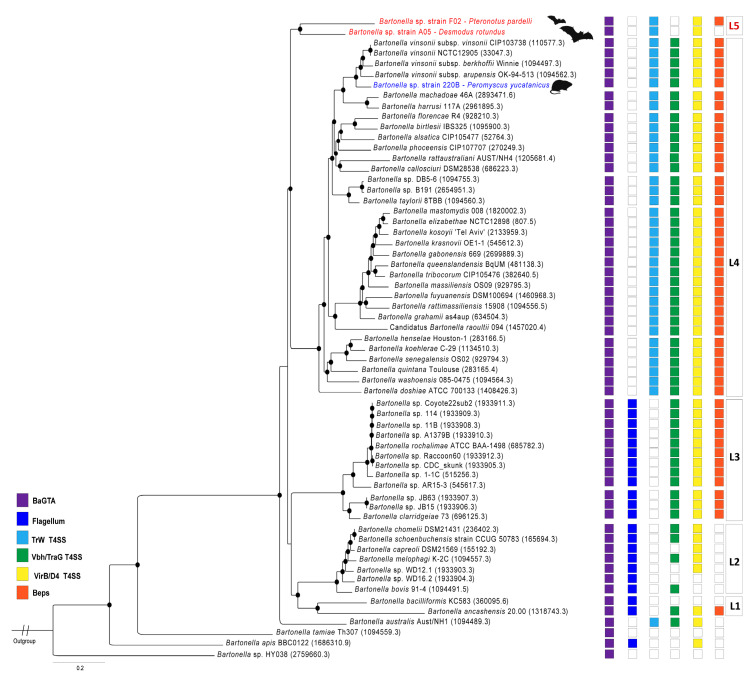
Whole-genome-based phylogeny. Phylogenetic tree of species of the genus *Bartonella* based on 149 shared protein genes and 137,193 nucleotide positions. The topology was reconstructed using the RAxML method on PATRIC database. Red branches represent the novel bat-associated *Bartonella* species. The blue branch represents the novel mouse-associated *Bartonella* species. Nodal support >95 of bootstrap values is shown as black circles. The genome ID references used in this analysis are described in brackets. The known *Bartonella* lineages are described on boxes on the right of main clades (L1–L4), the proposal of a new lineage is described in red (L5). The squares represent the virulence factors found in the *Bartonella* genomes: BaGTA (purple), Flagellum (sky blue), TrW T4SS (light blue), Vbh/TraG T4SS (green), Virb/D4 T4SS (yellow), and Beps (orange). Bar, substitutions per nucleotide position. *Brucella abortus* strain MC was used as an outgroup. Figure adapted from Wagner and Dehio [35].

**Figure 2 microorganisms-11-00340-f002:**
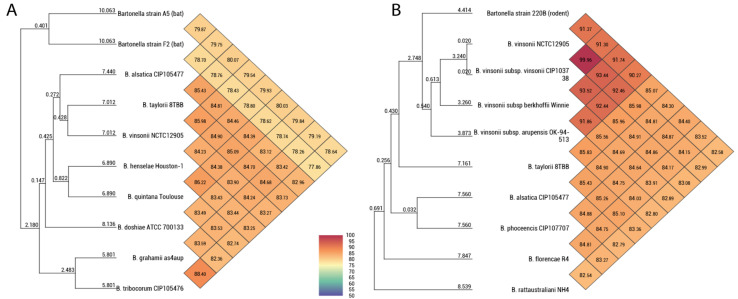
Heat map generated using OrthoANI values based on whole genome sequences, calculated using OAT software and other closely related species. (**A**) OrthoANI values of *Bartonella* strain A05 and strain F02. (**B**) OrthoANI values of *Bartonella* strain 220B. The numbers in each diamond at the diagonal lines correspond to the OrthoANI values between the species.

**Table 1 microorganisms-11-00340-t001:** Genome features of the three strains of *Bartonella* from one mouse and two bats.

	*Bartonella* Strain 220B	*Bartonella* Strain F02	*Bartonella* Strain A05
Host	*Peromyscus yucatanicus* (mouse)	*Pteronotus parnellii* (insectivorous bat)	*Desmodus rotundus* (vampire bat)
Contigs	67	20	19
Total length (bp)	1,929,784	1,448,328	1,559,146
DNA GC content (mol%)	38.46	36.20	38.49
Essential genes (106)	105	106	106
Completeness (%)	99.1	100	100
Contamination (%)	0.9	1.9	0.9
Quality (%)	94.6	90.5	95.5
Protein coding genes (CDS)	1978	1320	1424
tRNA genes	39	39	40
rRNA genes	3	3	3

## Data Availability

The data is published on NCBI by the Bioproject link: https://www.ncbi.nlm.nih.gov/bioproject/861808/.

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
