# Peer review of "Genomic Characterization of Three Novel *Bartonella* Strains in a Rodent and Two Bat Species from Mexico"

_microorganisms, 2023, doi:10.3390/microorganisms11020340_

Round 1

Reviewer 1 Report

This is a well written, clear, and straightforward presentation of 3 novel Bartonella genomes, isolated from blood of 2 different bat species and a Peromyscus mouse in different parts of Mexico. The background and interpretation appear sound; my area of expertise is not in sequencing, bioinformatics, or phylogenetics, so I cannot comment on the appropriateness of the analysis therein.

I have a few specific comments to strengthen this manuscript prior to publication:

Title:  I find the title to be a bit misleading, could you restate to make clear that these are isolates from one individual mouse and 2 individual bats?

Abstract: I think it would be beneficial to mention in the abstract the species of rodent and bats that the samples were obtained from. Also in line 25, indicating that this was whole genome sequencing, and that you also performed pangenome analysis.

Introduction: In the second paragraph, please clarify that you’re referring to genetic diversity (in the sense of genetic divergence between strains, rather than population level diversity). Your aim is clearly stated, but the background does not necessarily address why this is an important aim. Can you add a bit to explain why phylogenetic relationships and genome divergence for these 3 strains is important?

Conclusions: Can you comment on what is known about within-host genetic diversity of Bartonella? I understand this can’t be investigated with single isolates for WGS, but can you put the diversity seen between these novel strains in context of other studies that have investigated genetic diversity within individual hosts (or vectors) using other methods?

Author Response

Response to reviewer #1:

This is a well written, clear, and straightforward presentation of 3 novel Bartonella genomes, isolated from blood of 2 different bat species and a Peromyscus mouse in different parts of Mexico. The background and interpretation appear sound; my area of expertise is not in sequencing, bioinformatics, or phylogenetics, so I cannot comment on the appropriateness of the analysis therein.

Thank you.

I have a few specific comments to strengthen this manuscript prior to publication:

Title:  I find the title to be a bit misleading, could you restate to make clear that these are isolates from one individual mouse and 2 individual bats?

Whenever a new isolate is cultured and identified it is from an individual host. We accept partially the reviewer's comment in this regard. We however modified the title to represent our findings in a clearer way to meet the reviewer's comment. The title now reads:" Genomic characterization of three novel Bartonella strains in a rodent and two bat species from Mexico" The species of the rodent and bats were added to the abstract according to the reviewer's following comment.

Abstract: I think it would be beneficial to mention in the abstract the species of rodent and bats that the samples were obtained from.

The rodent and bat species were added to the abstract and so the location where they were captured, according to the reviewer's comment. Now it reads:" We describe here Bartonella isolates obtained from blood samples of one rodent (Peromyscus yucatanicus from San José Pibtuch, Yucatan) and two bat species (Desmodus rotundus from Progreso, and Pteronotus parnellii from Chamela-Cuitzmala) from Mexico. (lines 23-24). We also corrected at the lines 60-61.

Also in line 25, indicating that this was whole genome sequencing, and that you also performed pangenome analysis.

We added the pangenome analysis to the abstract as requested by the reviewer (lines 25-26). There is no need to add whole genome sequencing (WGS) as we used the word "genomic" which eventually relates to WGS.

Introduction: In the second paragraph, please clarify that you’re referring to genetic diversity (in the sense of genetic divergence between strains, rather than population level diversity).

In line 50 we added the word "genomic" before the word diversity, as requested by the reviewer. We added the word genomic rather than genetic as most strains reported from the Americas were characterized at the genetic level (mainly genotypes) rather than genomic level (WGS).

Your aim is clearly stated, but the background does not necessarily address why this is an important aim. Can you add a bit to explain why phylogenetic relationships and genome divergence for these 3 strains is important?

As expressed in the abstract (lines 42-46): " As the greatest diversity of Bartonella species is reported in rodents and bats (Gutierrez et al., 2015; Stuckey et al., 2017; Han et al., 2021), rodents and possibly bats are considered a potential source of emerging zoonotic Bartonella pathogens (Krügel et al., 2022). Moreover, bats were reported to have a crucial role in the diversification of the Bartonella genus (McKee et al., 2021)", these sentences explain exactly why it is important, and answers the reviewer's question. Additionally, the first two lines of the conclusions section (which were modified according to the suggestions of reviewer #2) also elaborate on this point: " A deeper understanding of the evolutionary diversification seen in this genus is made possible by the characterization and description of additional genomes in rodents and bats."

Conclusions: Can you comment on what is known about within-host genetic diversity of Bartonella? I understand this can’t be investigated with single isolates for WGS, but can you put the diversity seen between these novel strains in context of other studies that have investigated genetic diversity within individual hosts (or vectors) using other methods?

This point is a very important one but could not be explored in our current study as we found one Bartonella species per one host (and all hosts were different). As within-host genetic diversity was not part of our study we find no place to mention it in the conclusion section. We however added the words "genomic approach" (line 273) to the first sentence of the conclusions. We further wrote in the original text (line 275) that we used "phylogenomic clustering and virulence factors" analyses as compared to previous studies that used mainly genetic rather than genomic characterization. We therefore concentrated in the conclusions on the main findings of our current study.

Reviewer 2 Report

Review Reports

Dear Editor in Chief

Article Title: Genomic characterization of three novel Bartonella strains in 2 rodents and bats from Mexico

General comments

Gram-negative, facultative intracellular bacteria called Bartonella species infect erythrocytes and endothelium cells in mammals and are spread by arthropods. There are currently three subspecies and 39 species with recognized taxonomic status, at least 20 of which have been linked to human diseases. Rodents and  bats are thought to be a possible source of newly zoonotic Bartonella diseases since rodents and bats have the highest recorded diversity of Bartonella species. A deeper understanding of the evolutionary diversification seen in this genus is made possible by the characterization and description of additional Bartonella genomes in rodents and bats. Novel Bartonella species described that need further characterization in order to be fully named according to the ICNP. Based on phylogenomic clustering and virulence factors, the bat-associated Bartonella species, presented in this study, suggest the possibility of a new fifth lineage in the genus.

Please make highlighted any new changes

Abbreviation, please indicate full words for first use at the text of article.

References should be based on journal constriction.

# Reviewer comment #

§   Introduction please mention problem

§  METHODS ethical approval needs

§  Results please insert PCR gel electrophoresis  Figures

§  Please make highlighted any new changes

§  Abbreviation, please indicate full words for first use at the text of article.

§  Update the references very carefully.

§  References should be based on journal constriction

Author Response

Response to reviewer #2:

General comments

Gram-negative, facultative intracellular bacteria called Bartonella species infect erythrocytes and endothelium cells in mammals and are spread by arthropods. There are currently three subspecies and 39 species with recognized taxonomic status, at least 20 of which have been linked to human diseases. Rodents and bats are thought to be a possible source of newly zoonotic Bartonella diseases since rodents and bats have the highest recorded diversity of Bartonella species. A deeper understanding of the evolutionary diversification seen in this genus is made possible by the characterization and description of additional Bartonella genomes in rodents and bats. Novel Bartonella species described that need further characterization in order to be fully named according to the ICNP. Based on phylogenomic clustering and virulence factors, the bat-associated Bartonella species, presented in this study, suggest the possibility of a new fifth lineage in the genus.

Thank you for suggesting better phrasings which we fully agree with – we therefore rephrased the text according to the reviewer's suggestions. Numbers of species and writing style were corrected accordingly - see highlighted lines 40-41, 50-51 and 271-3.

# Reviewer comment #

  • Introduction please mention problem

We elucidated the problem that reads now: " …..the genomic diversity of Bartonella species detected or isolated in mammal hosts, especially in the Americas has not been fully described and has yet to be investigated and elucidated (Stuckey et al., 2018; Schulte-Fischedick et al., 2016). – lines 49-52.

  • METHODS ethical approval needs

We used samples from previously published studies (Shulte-Fishedick et al., 2016; Stuckey et al., 2018) – both have IACUC (ethics) approvals indicated in the original papers that were cited in this current manuscript.

  • Results Please inserted PCR gel  electrophoresis  Figures

As our whole genome raw data was deposited in GenBank (as part of 100K genomic project at UC Davis, CA, USA) and now is public and can be analyzed and scrutinized by the readers for lengths, contaminations, assembly, etc… we see no reason to add the PCR gel figures (Lines 122-126 in our manuscript – "This Whole Genome Shotgun project has been deposited at DDB/ENA/GenBank under accession numbers: JANHOI000000000 (strain 220B); JANHOJ000000000 (strain A05); JANHOK000000000 (strain F02); Bioproject PRJNA8621808; BioSample: SAMN29927589- SAMN29927591").

Please make highlighted any new changes

Done - All changes in the text were highlighted as requested

  • Abbreviation, please indicate full words for first use at the text of article.

  • Update the references very carefully. Thank you very much for noticing this. Done.

  • References should be based on journal constriction

DONE!

Reference Schulte-Fischedick et al 2016 was corrected.

Reference Titus Brown et al 2016 was corrected to Brown et al.2016.

Reference La Scola et al 2005 was corrected to La Scola et al 2003 and added to the reference list.

Added references:

Amaral, R. B., Cardozo, M. V., Varani, A. de M., Furquim, M. E. C., Dias, C. M., Assis, W. O. de, da Silva, A. R., Herrera, H. M., Machado, R. Z., & André, M. R. (2022). First Report of Bartonella spp. in Marsupials from Brazil, with a Description of Bartonella harrusi sp. nov. and a New Proposal for the Taxonomic Reclassification of Species of the Genus Bartonella. Microorganisms, 10(8), 1609. https://doi.org/10.3390/microorganisms10081609

Bai, Y., Kosoy, M., Recuenco, S., Alvarez, D., Moran, D., Turmelle, A., Ellison, J., Garcia, D. L., Estevez, A., Lindblade, K., & Rupprecht, C. (2011). Bartonella spp. in bats, Guatemala. Emerging Infectious Diseases, 17(7), 1269–1272. https://doi.org/10.3201/eid1707.101867

Becker, D. J., Bergner, L. M., Bentz, A. B., Orton, R. J., Altizer, S., & Streicker, D. G. (2018). Genetic diversity, infection prevalence, and possible transmission routes of Bartonella spp. in vampire bats. PLoS Neglected Tropical Diseases, 12(9), 1–21. https://doi.org/10.1371/journal.pntd.0006786

Benavides, J. A., Valderrama, W., Recuenco, S., Uieda, W., Suzán, G., Avila-Flores, R., Velasco-Villa, A., Almeida, M., de Andrade, F. A. G., Molina-Flores, B., Vigilato, M. A. N., Pompei, J. C. A., Tizzani, P., Carrera, J. E., Ibanez, D., & Streicker, D. G. (2020). Defining New Pathways to Manage the Ongoing Emergence of Bat Rabies in Latin America. Viruses, 12(9), 1–13. https://doi.org/10.3390/v12091002

Brettin, T., Davis, J. J., Disz, T., Edwards, R. A., Gerdes, S., Olsen, G. J., Olson, R., Overbeek, R., Parrello, B., Pusch, G. D., Shukla, M., Thomason, J. A., Stevens, R., Vonstein, V., Wattam, A. R., & Xia, F. (2015). RASTtk: A modular and extensible implementation of the RAST algorithm for building custom annotation pipelines and annotating batches of genomes. Scientific Reports, 5. https://doi.org/10.1038/srep08365

Buffet, J., Kosoy, M., & Vayssier-Taussat, M. (2013). Natural history of Bartonella -infecting rodents in light of new knowledge on genomics, diversity and evolution. Future Microbiology, 8(9), 1117–1128. https://doi.org/10.2217/fmb.13.77

Johnson, N., Aréchiga-Ceballos, N., & Aguilar-Setien, A. (2014). Vampire bat rabies: Ecology, epidemiology and control. Viruses, 6(5), 1911–1928. https://doi.org/10.3390/v6051911

La Scola, B., Zeaiter, Z., Khamis, A., & Raoult, D. (2003). Gene-sequence-based criteria for species definition in bacteriology: The Bartonella paradigm. Trends in Microbiology, 11(7), 318–321. https://doi.org/10.1016/S0966-842X(03)00143-4

Schwengers, O., Jelonek, L., Dieckmann, M. A., Beyvers, S., Blom, J., & Goesmann, A. (2021). Bakta: Rapid and standardized annotation of bacterial genomes via alignment-free sequence identification. Microbial Genomics, 7(11). https://doi.org/10.1099/MGEN.0.000685

Stuckey, M. J., Chomel, B. B., de Fleurieu, E. C., Aguilar-Setién, A., Boulouis, H. J., & Chang, C. chin. (2017). Bartonella, bats and bugs: A review. Comparative Immunology, Microbiology and Infectious Diseases, 55(September), 20–29. https://doi.org/10.1016/j.cimid.2017.09.001

Stuckey, M. J., Chomel, B. B., Galvez-Romero, G., Olave-Leyva, J. I., Obregón-Morales, C., Moreno-Sandoval, H., Aréchiga-Ceballos, N., Salas-Rojas, M., & Aguilar-Setién, A. (2017). Bartonella infection in hematophagous, insectivorous, and phytophagous bat populations of Central Mexico and the Yucatan Peninsula. American Journal of Tropical Medicine and Hygiene, 97(2), 413–422. https://doi.org/10.4269/ajtmh.16-0680

Seemann, T. (2014). Prokka: Rapid prokaryotic genome annotation. Bioinformatics, 30(14), 2068–2069. https://doi.org/10.1093/bioinformatics/btu153

Tatusova, T., Dicuccio, M., Badretdin, A., Chetvernin, V., Nawrocki, E. P., Zaslavsky, L., Lomsadze, A., Pruitt, K. D., Borodovsky, M., & Ostell, J. (2016). NCBI prokaryotic genome annotation pipeline. Nucleic Acids Research, 44(14), 6614–6624. https://doi.org/10.1093/nar/gkw569

Excluded reference:

Mediannikov, O., Aubadie, M., Bassene, H., Diatta, G., Granjon, L., & Fenollar, F. (2014). Three new Bartonella species from rodents in Senegal. International Journal of Infectious Diseases, 21, 335. https://doi.org/10.1016/j.ijid.2014.03.1112   
